# Hatching-Box: Automated in situ monitoring of *Drosophila melanogaster* development in standard rearing vials

**Julian Bigge**[1,2], **Maite Ogueta**[3*], **Luis Garcia**[1,3], **Benjamin Risse**[1,2*]

**1** Institute for Geoinformatics, University of Münster, Münster, Germany, **2** Faculty of Mathematics and Computer Science, University of Münster, Münster, Germany, **3** Institute of Neuro- and Behavioral Biology, University of Münster, Münster, Germany

* m.ogueta@uni-muenster.de; b.risse@uni-muenster.de

**Data availability statement:** The data used in this study is publicly available at

## Abstract

In this paper we propose the Hatching-Box, a novel in situ imaging and analysis system to automatically monitor and quantify the developmental behavior of *Drosophila melanogaster* in standard rearing vials and during regular rearing routines, reducing the need for explicit experiments.This is achieved by combining custom tailored imaging hardware with dedicated detection and tracking algorithms, enabling the quantification of larvae, filled/empty pupae and flies over multiple days. Given the affordable and reproducible design of the Hatching-Box in combination with our generic client/server-based software, the system can easily be scaled to monitor an arbitrary amount of rearing vials simultaneously. We evaluated our system on a curated image dataset comprising nearly 416,000 annotated objects and performed several studies on real world experiments. We successfully reproduced results from well-established circadian experiments by comparing the eclosion periods of wild type flies to the clock mutants *per^short^*, *per^long^* and *per^0^* without involvement of any manual labor. Furthermore we show, that the Hatching-Box is able to extract additional information about group behavior as well as population development and activity. These results not only demonstrate the applicability of our system for long-term experiments but also indicate its benefits for automated monitoring in the general cultivation process.

## Introduction

The model organism *D. melanogaster* has been used to study different aspects of biology, such as genetics, neuroscience or cell biology, and in recent years even for translational studies for human diseases [1,2]. The whole life cycle of *D. melanogaster* comprises 4 different stages: egg, larvae, pupae and adults, and often it is of interest to know when and how many animals enter a specific developmental stage and how long they do remain in this stage. One commonly studied transition is the adult emergence or eclosion of *D. melanogaster*. In 1971, Konopka and Benzer found that this is time of day dependent and that mutations in a single gene named *period* affected this process [3]. The mutants they used do not only show a change on the timing of eclosion, but also show diverging pace in their development as a whole [4].

https://doi.org/10.17879/53998663190 and the source code can be found https://zivgitlab.uni-muenster.de/j_bigg01/Hatch.

**Funding:** The author(s) received no specific funding for this work.

**Competing interests:** The authors have declared that no competing interests exist.

The study of eclosion is still a common procedure to study the circadian rhythms of the flies [5], as well as the measurement of the timing of pupation as an indicator for the development [6]. This type of experiments are time-consuming and often involve the constant personal monitoring of the rearing vials. Despite the growing necessity in various research domains, circadian experiments continue to be a challenging and time-consuming task to this day since the tools available either require fluorescent dyes to visually mark interesting brain regions [7] or extracting individual animals from their rearing vials and monitoring them in a distinctive system. An example of such a system is the commonly used Trikinetics Eclosion Monitor for which *D. melanogaster* pupae are glued to a disc and emerging adult flies fall down and are counted as they cross an infrared barrier [8]. Other, camera-based systems, record videos of the animals which can then be annotated semi-automatically by the experimenter with an imaging software such as Fiji [9–11]. For more detailed studies, automated tracking systems have been developed which automatically extract the behavior of individual flies or larvae over time and were surveyed by Panadeiro et al. [12]. For example anTrax, Ctrax and Idtracker.ai are commonly used tracking applications that allow tracking of multiple *D. melanogaster* over a period of hours [13–15]. Providing a fine-tuned combination of a custom made imaging system and tracking software, Risse et al. proposed FIMTrack [16]. The proposed imaging hardware consists of an arena made from an acrylic glass plate which is illuminated by infrared LEDs using frustrated total internal reflection (FTIR) and recorded with an infrared sensitive camera. This system was extended to be used for *D. melanogaster* larvae crawling in FTIR-illuminated rearing vials, yielding fine-grained 3-dimensional trajectories of the animals' movement [17]. Similarly for the domain of ethology Geissmann et al. designed an open source ethoscope system which can track adult flies [18] and analyze their behavior. Apart from methods focused on *D. melanogaster* there is also DeepLabCut which is often used for mice [19] but is also applicable for other animals including flies.

The aforementioned systems share considerable limitations and are often not directly applicable for long term experiments and lifelong monitoring as needed in circadian rhythm and other experiments. For example, anTrax, Ctrax, IdTracker.ai and DeepLabCut do not include specific imaging hardware which can cause difficulties in applying these software to custom imaging conditions and experimental setups. Moreover, identifying a suitable combination of cameras and recording software can have a significant influence on the performance of the tracking algorithms. Methods such as FIMTrack or the ethoscope combine software and adapted imaging hardware but still rely on explicitly prepared and implemented experiments. The same also holds for the commercially available monitoring systems from Trikinetics, which only yields very basic activity information of the animals.

Another substantial constraint of these methods with regards to long-term behavior experiments is their inability to discriminate between different developmental stages of *D. melanogaster* which is a prerequisite to study the life cycle of flies. Since 2023, flyGear [20] offers a commercially available system that can be used to classify D. melanogaster larvae and pupae, counts them and provides automatic data analysis and visualization.

To enable high-throughput, long-term behavior experiments as well as automatic monitoring of *D. melanogaster* development during breeding, we propose an open-source system called *Hatching-Box*, intended to provide an optimal trade-off between hardware complexity, usability and result quality. Our system provides numerous advantages compared to current state-of-the-art:

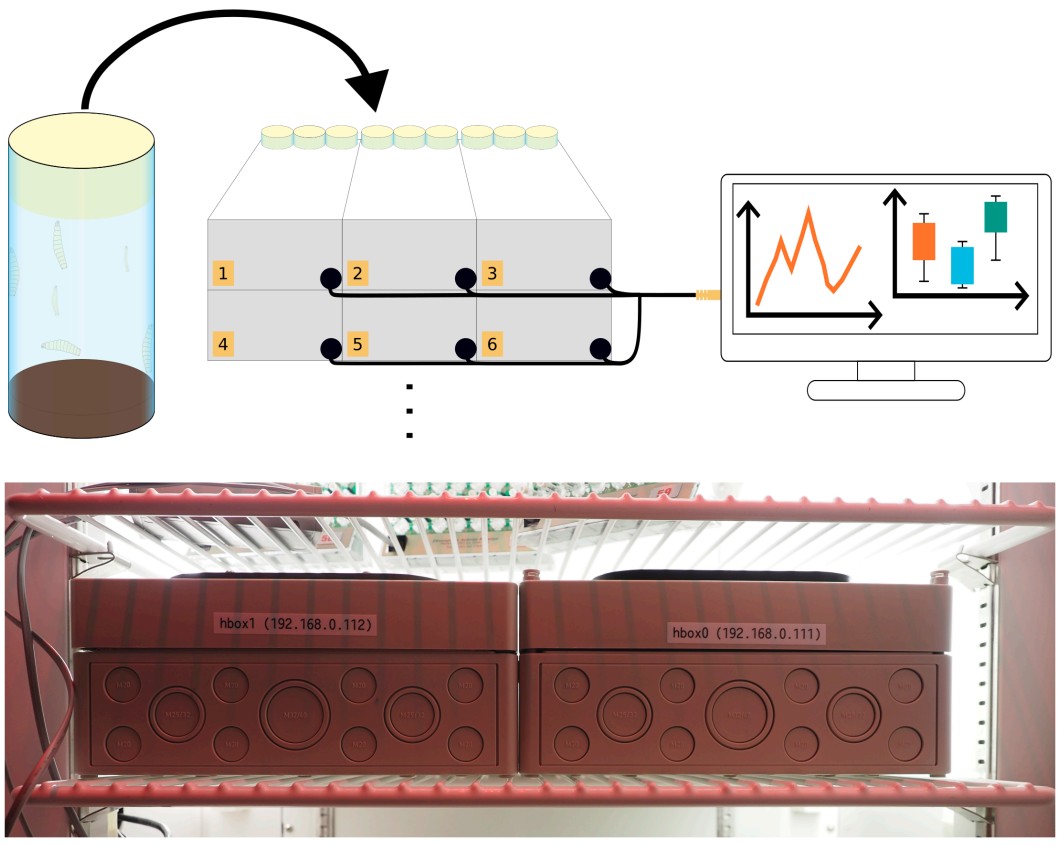

**Fig 1. System overview showing stackable Hatching-Boxes.** Standardized rearing vials used to house *D. melanogaster* can be placed in the Hatching-Box, which then automatically tracks the population's behavior and provides images and behavior analysis to a central computer.

1. Life-time monitoring of *D. melanogaster* is facilitated in off-the-shelf rearing vials during the conventional rearing routine. As a consequence, our system does not require the manual collection of animals and no preparation of the tracking system is necessary, enabling behavioral quantifications without any labor overhead.

2. Each system houses up to three rearing vials (41.5mm diameter), can be placed in incubators or cultivation rooms and comprises primarily off-the-shelf hardware components with some 3D printed and manufactured parts from easily obtainable materials so that an arbitrary amount of stackable monitoring systems can run in parallel (see Fig 1).

3. The self-contained hardware design renders the internal imaging modalities independent from external influences. Hence, the hardware, firmware and software are precisely aligned to fit this setup to yield accurate and reproducible results.

4. A machine learning-based object detection algorithm based on the YOLO object detector is adapted and trained on nearly 416.000 manually labeled objects. Our system is capable of accurately detecting all developmental stages of *D. melanogaster* even in very cluttered conditions, yielding an accuracy of up to 91% while providing real-time processing capabilities.

5. An array of built-in sensors is integrated to monitor the temperature and humidity within each box. In addition, dedicated light stimulation has been integrated, which can

either automatically adapt the inner lightning conditions to the external brightness or can be programmed to provide individual illumination schemes for each box.

6. We implemented an easy-to-use GUI that can be used for controlling the camera, start or stop a recording and automatically analyze the data.

7. All components (i.e. hardware, firmware, software) have been tested and a rigorous evaluation of our system was conducted by reproducing the results of Konopka and Benzer in a well-established circadian experiment by using the three *D. melanogaster* clock mutants $per^{short}$, $per^{long}$ and $per^0$ and comparing their activity cycles with the wild type.

8. The full source code, collected dataset, a conclusive bill of materials and required files for the 3D printed and manufactured parts will be made available under an open-source license.

## Results

### Hardware

The hardware of our system is designed to be easily integrated into established daily laboratory routines to provide long-term monitoring capabilities without the need for additional manual labor. This requires the use of rearing vials for animal housing and a small overall footprint of the system to fit into most incubators. While we use rearing vials with a 41.5mm diameter, our system can be easily adapted to other vials as well by editing one of the provided parametric manufacturing files (see S1 File). Additional sensors and light sources provide supervision and control over the breeding conditions in each box. We use a standard small-sized plastic electrical box for the main compartment, housing our central computing device, a Raspberry Pi 4, a camera and additional necessary technical components and up to three standard rearing vials (see Fig 2). The form factor of these boxes enable arbitrary stacking of multiple setups for high-throughput experiments and each box is individually connected via Ethernet to operate several systems in parallel. The built in camera captures frames using a user-specified frame rate allowing for continuous recordings or time-lapse sequences. For illumination an acrylic based light guide panel is positioned behind the rearing vials which is equipped with several near infrared and white light-emitting diodes (LEDs). In order to visually detect small and semi-translucent objects (e.g. empty pupae, larvae, etc.) we optimized the transmitted light configuration by deriving a custom diffusion pattern, providing homogeneous diffuse illumination across the light guide panel (see S1 Appendix). The infrared (IR) LEDs are used for image capturing only given this wavelength is reported to be invisible for flies [21]. An additional IR shield in the middle of the compartment prevents light emitted by the computing hardware to have an influence on the specimens or produce reflections on the rearing vials. We synchronized the IR LED activation with the camera so that IR light is only present during image acquisition while being turned off otherwise to prevent heat buildup and to keep a low energy profile . Temperature and humidity sensors inside each box record temporally synchronized measures of the breeding conditions. The white LEDs provide a visual stimuli and can either be freely programmed for each box individually or can be synchronized with the outside brightness using an externally mounted brightness sensor (e.g. to mimic the lightning conditions of the surrounding incubator). The combination of the high resolution camera and the optimized backlight panel yield detailed images of all developmental stages of *D. melanogaster* (see Fig 2c) which are subsequently used for data collection and in our classification and tracking pipeline.

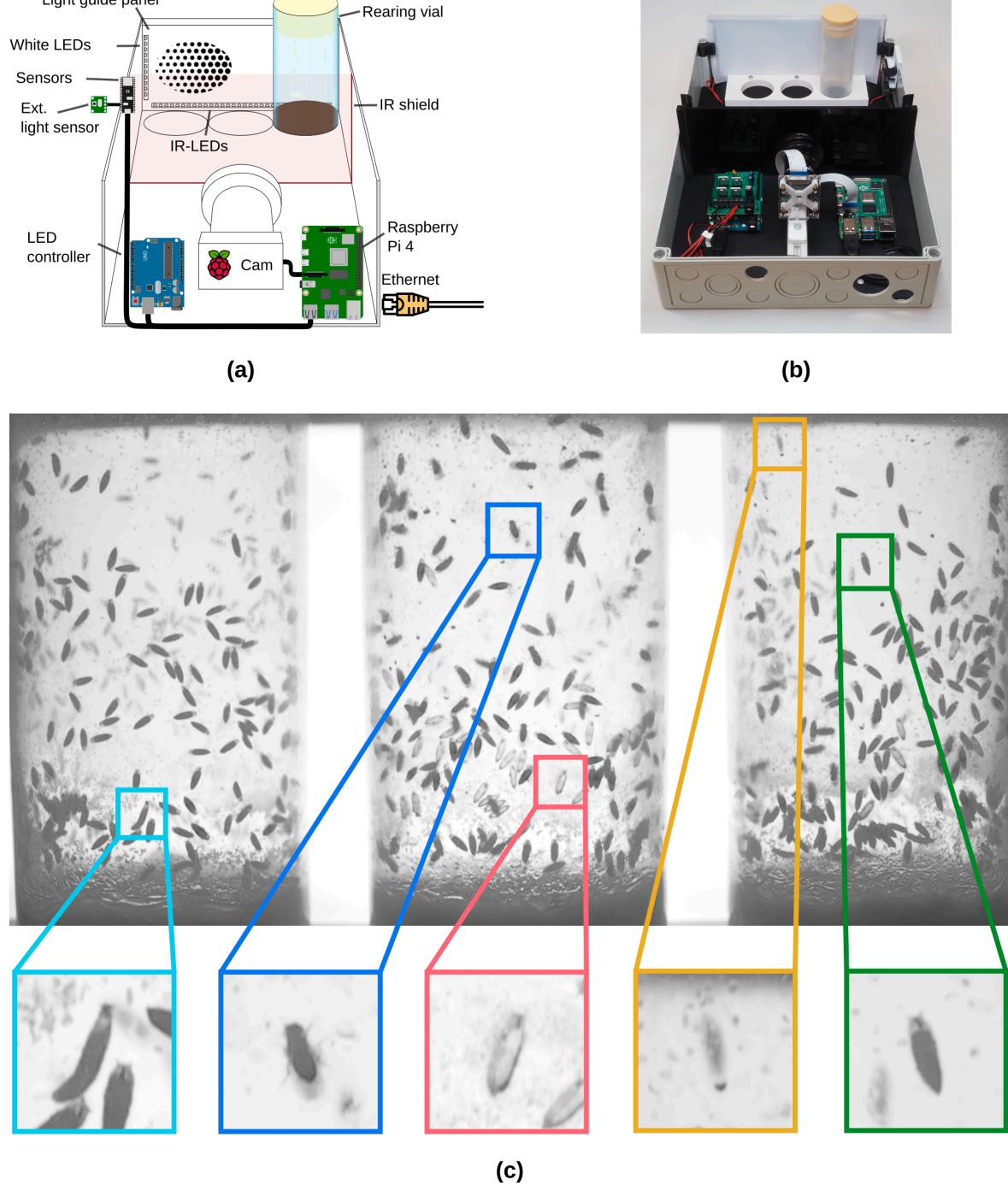

**Fig 2. Overview of the proposed hardware architecture.** (a) A single Hatching-Box consists of a RaspberryPi 4 with the HQ camera module, an LED controller device (Arduino Uno + custom shield), an Arduino Sense BLE 33 sensor board, as well as a light guide panel, outfitted with IR and white LEDs. (b) Picture of a Hatching-Box with cover removed. (c) Image as taken by our Hatching-Box with different objects highlighted: third instar larva (cyan), adult fly (blue), empty pupa (red), out-of-focus (yellow) and full pupa (green).

## Dataset

For training our YOLOv7 object detector we curated a dataset containing *D. melanogaster* in various developmental stages, i.e. *larva*, *pupa*, *empty pupa*, *adult fly*. For the *larva* class, only wandering third instar larvae were labelled as L1 and L2 larvae stay in the food at the bottom of the vial and are only occasionally visible in our recordings. In addition we marked out-of-focus objects on the backside of the vials since these blurred objects can be the cause of misdetections and misclassifications. In total we annotated 415,709 objects (bounding boxes and class labels) in 1348 images recorded in different Hatching-Box systems. The first 200 images were labelled thoroughly by domain experts. We utilized these first images to train an initial YOLOv7-E6E model which was subsequently used to provide suggestions to the annotators to speed up the labeling process. Importantly, all suggestions proposed by our model were manually checked by domain experts. Table 1b shows the class distribution across our dataset. The dataset can be accessed at https://doi.org/10.17879/53998663190.

## Software

**Detection and classification of *D. melanogaster* using YOLOv7.** Detection and classification (distinguishing different developmental stages of *D. melanogaster*) in the images captured by our system is performed by the YOLOv7 machine learning model [22]. YOLOv7 provides various versions, i.e. the base model, -tiny, -X, -E6, -W6 and -E6E, comprising different depth of scaling pyramids helping to detect smaller objects in exchange for higher parameter count and compute overhead. We trained these models on our curated dataset with the same train-validation split (80:20) for 300 epochs and compared their performance

**Table 1**. **Performance analysis of the trained YOLOv7 models.** 1a Comparison of average precision (AP), average recall (AR) and mean average precision (mAP) of the YOLOv7 models trained on our Hatching-Box dataset (out-of-focus objects excluded). 1b Overview of the class distribution of our curated dataset. 1c Average inference and tracking time comparison in ms/frame on a CPU (AMD Ryzen 7 3700X 8-Core) and GPU (NVIDIA GeForce RTX 3060 Ti).

| Model | $AP^{val}$ | $AR^{val}$ | $mAP^{val}_{50}$ | $mAP^{val}_{50:95}$ | Class | Count | Share |
|---|---|---|---|---|---|---|---|
| YOLOv7-tiny | 91.93% | 90.01% | 93.28% | 73.03% | Empty Pupa | 133,022 | 32.00 % |
| YOLOv7-X | 91.29% | 89.84% | 92.91% | 75.39% | Out-of-focus | 127,511 | 30.67 % |
| YOLOv7 | 91.65% | 90.27% | 93.23% | 74.87% | Full Pupa | 91,617 | 22.04 % |
| YOLOv7-E6 | 91.69% | 90.06% | 93.62% | 77.16% | Adult Fly | 41,445 | 9.97 % |
| YOLOv7-W6 | 91.23% | 89.62% | 94.01% | 77.38% | Larva | 22,114 | 5.32 % |
| YOLOv7-E6E | 90.55% | 90.05% | 93.27% | 76.35% | **Total** | 415,709 | |
| (a) | | | | | (b) | | |

| | Model | Inference | | Tracking | | Overall | |
|---|---|---|---|---|---|---|---|
| **CPU** | YOLOv7-tiny | 438.69 | (±21.94) | 132.79 | (±19.27) | 611.31 | (±27.63) |
| | YOLOv7-X | 5240.78 | (±90.83) | 112.28 | (±15.03) | 5393.07 | (±91.19) |
| | YOLOv7 | 3162.86 | (±70.04) | 120.23 | (±16.02) | 3323.05 | (±69.99) |
| | YOLOv7-E6 | 3639.89 | (±79.95) | 132.93 | (±19.61) | 3812.84 | (±80.40) |
| | YOLOv7-W6 | 2328.11 | (±59.51) | 129.02 | (±18.09) | 2497.16 | (±58.69) |
| | YOLOv7-E6E | 5718.86 | (±104.26) | 136.52 | (±21.87) | 5895.45 | (±104.78) |
| **GPU** | YOLOv7-tiny | 133.12 | (±4.62) | 160.87 | (±17.33) | 335.72 | (±16.86) |
| | YOLOv7-X | 336.64 | (±6.19) | 135.90 | (±12.56) | 514.65 | (±12.47) |
| | YOLOv7 | 243.77 | (±4.76) | 144.87 | (±13.09) | 430.83 | (±13.24) |
| | YOLOv7-E6 | 271.81 | (±6.20) | 162.68 | (±17.25) | 476.60 | (±16.76) |
| | YOLOv7-W6 | 211.72 | (±4.02) | 157.63 | (±15.41) | 411.47 | (±15.47) |
| | YOLOv7-E6E | 370.27 | (±6.13) | 165.77 | (±20.87) | 578.25 | (±19.98) |
| (c) | | | | | | | |

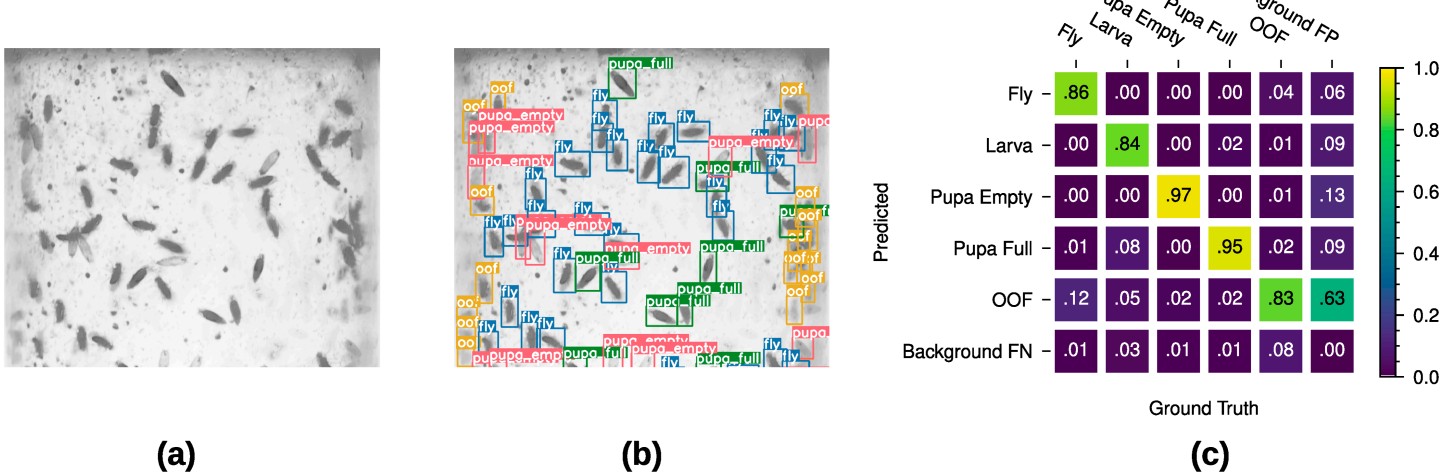

**Fig 3. Class confusion evaluation for the trained YOLOv7-tiny model.** (a), (b) Crop of an image as taken with our system before and after object detection by YOLOv7. The detected objects are third instar larvae (cyan), adult flies (blue), empty pupae (red), full pupae (green) and out-of-focus (yellow). (c) Class confusion matrix of the used YOLOv7-tiny model. (For full comparison see S1 Fig).

on the validation dataset. In our experiments we found that for the task of detection and classification of *D. melanogaster* in our Hatching-Box images, the different model variants show comparable performance with regards to their classification accuracy. The best performing model shows 4.35% better mean average precision (mAP) on 50% to 95% intersection over union (IoU) compared to our YOLOv7-tiny baseline, while it only shows a surplus of 0.73% in mAP on 50% IoU (see Fig 1). However, deep scaling pyramids as supplied by the larger YOLOv7 models (E6, W6, E6E) are not shown to be beneficial for overall classification performance. Additionally, we can observe a small decrease in average precision (AP) compared to the other models which we trace back to slight overfitting on our training data.

We also evaluated class confusion for our model to evaluate its ability in recognizing different *D. melanogaster* developmental stages (Fig 3). Specifically, full and empty pupae are detected with a 95% and 97% accuracy and are only occasionally confused with each other. Adult *D. melanogaster* are correctly detected in 86% of the cases, in 12% of the cases the fly is detected incorrectly as an out-of-focus object which can be explained by their ability to freely move in the vial compared to larvae and pupae, escaping the focal plane of the camera. Similarly, detection of larvae shows a 84% accuracy, most commonly confused with full pupae (8%), since these two classes have a similar appearance in single channel grayscale IR images. As can be seen in the confusion matrix our models successfully differentiates between *D. melanogaster* and environmental clutter: only 1% of adult flies and (full/empty) pupae and only 3% of larvae are misclassified as background (background false negatives). Conversely, in the case a background object is detected as a foreground object (background false positives), it is discarded as the out-of-focus class in 63% of the time. Over the course of the performed experiments, our detection algorithm identified up to 500 unique specimens per image. As a direct result of our high-throughput system, single misdetections on a frame-by-frame basis can statistically be compensated over the course of the whole experiment. We achieve additional robustness for the detections by implementing a temporal association mechanism which takes multiple previous reference time points into account to produce a most probable identity matching over a whole image sequence.

**Identity preserving tracking.** As mentioned above, YOLOv7 provides detections (bounding boxes) surrounding the foreground objects and associated class labels, namely adult fly, filled pupae, empty pupae, larvae and out-of-focus object. The former four object types are subsequently used in an identity preserving tracking algorithm to compute continuous trajectories for all individuals over time . By jointly considering the bounding box location and area as well as the object class, temporal association of a box at time $t$ and $t-1$ requires an IoU of more than 60% and the same or adjacent developmental stages between two consecutive frames. In addition, temporal smoothing is used by integrating multiple past frames which also enables the preservation of identities in case of occasionally missed detections (e.g. due to object occlusions). For our scenario a time window of three additional frames at $t-2$, $t-3$ and $t-4$ yielded good results.

During our circadian experiment we have captured with a framerate of one frame each 10 minutes. This approach effectively reduces the amount of accumulated data while still allowing us to monitor group behavior and spatial distribution (see S3 Fig). At the same time, this temporal resolution is sufficient to detect pupation and eclosion events in the lifecycle of *D. melanogaster* used as indicator for their circadian rhythm, as the specimens stay immobile during the pupal stage and therefore do not move inbetween frames. Since the behavioral and morphological changes associated with pupation are more gradual and often rely on indirect proxies (such as periods of immobility), automatically determining the precise timing of pupation events is inherently less accurate than for binary eclosion events (see S4 Fig).

Additionally, as a proof-of-concept, we demonstrated that our system is capable of tracking larval locomotion by recording at the current maximum framerate of 1 fps (see Fig 4). This preliminary test illustrates the system's potential applicability for behavioural analysis of wandering L3 larvae, even though the framerate is presently limited. To address the limitation that larvae may move out of the focal plane and become difficult to track in single-camera systems, our detection model is trained to recognize out-of-focus larvae as a separate class. This enables us to maintain accurate population counts by tracking both in-focus and out-of-focus objects, while avoiding explicit classification into pupae or larvae when image quality is insufficient. In contrast, tracking of fast moving adult flies would require a significantly higher framerate which was not tested during the course of the experiment, albeit technically supported.

Life-long monitoring and complex experimental settings require recordings over multiple weeks resulting in thousands of trajectories. However, these also include trajectories that only cover a small number of frames, usually occurring for adult *D. melanogaster* that can move particularly fast when flying and hence cannot be tracked using the aforementioned frame rate. Given our focus on the transitions in developmental stages we discarded small trajectories below 30 consecutive detections.

To assess the performance of our tracking pipeline we measured the average computation required for tracking a random sequence of 100 consecutive frames of a video of ten runs with each of the previously introduced YOLOv7 models as a detector (see Fig 1). We compare the time the YOLOv7 model needs for inference and the time required for temporal association as introduced in this section on an AMD Ryzen 7 3700X 8-Core CPU and a NVIDIA GeForce RTX 3060 Ti GPU. On average, each frame of the selected random sequence has around 500 detected objects (out-of-focus objects excluded) that have to be associated between consecutive frames. Nevertheless, when run on the CPU, the average overhead of our tracking algorithm compared with only inference of the YOLOv7 models was between 112.28 and 136.52 ms. For the basic, less resource intensive YOLOv7-tiny, the temporal association algorithm on average constitutes 24.7% of the overall runtime per frame. When used with the most complex YOLOv7 model, YOLOv7-E6E, the share of total runtime decreases to only 2.3%.

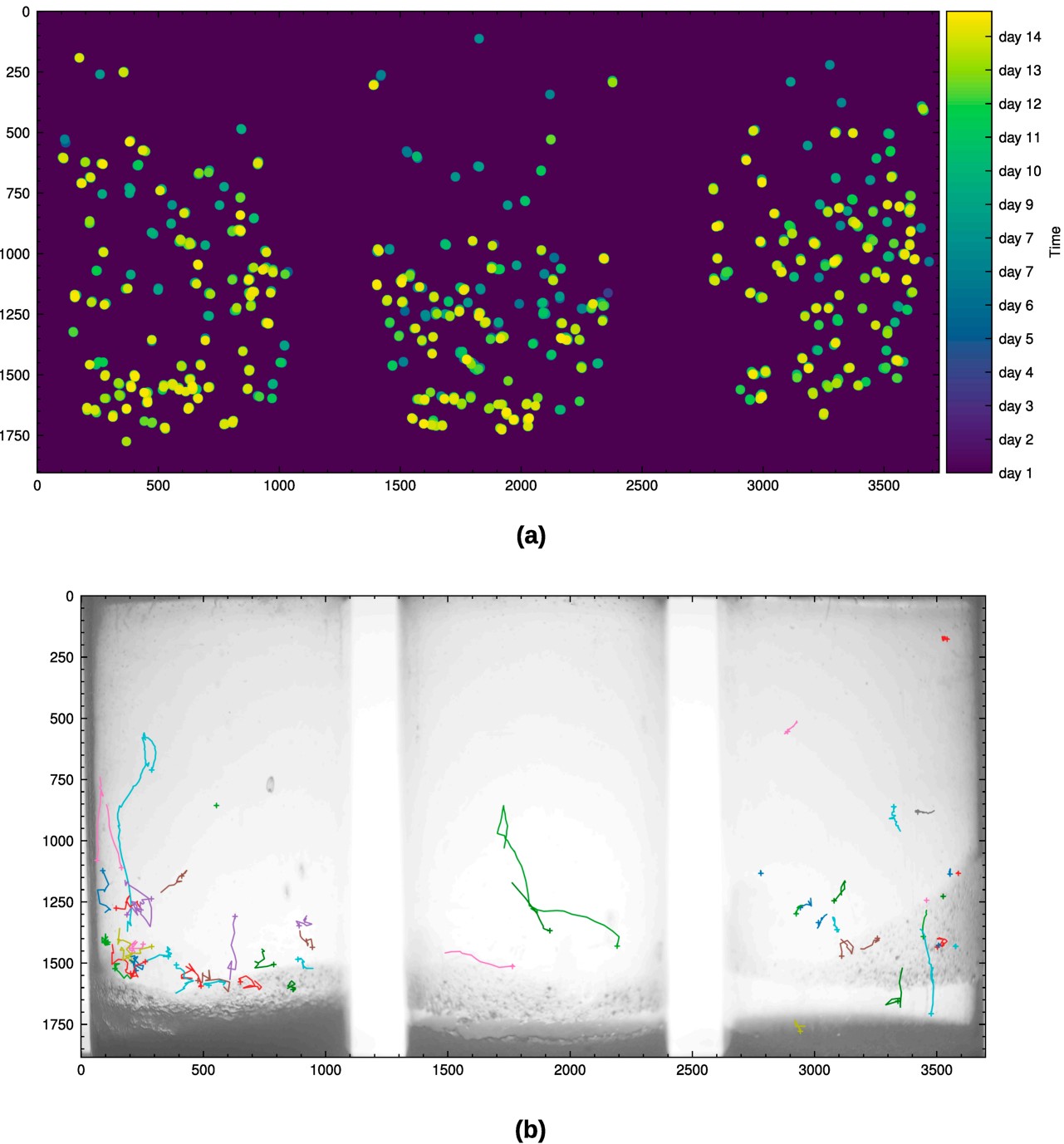

**Fig 4. Performance analysis for identity preserving tracking.** (a) Positions of eclosion (wildtype *D. melanogaster*) over the course of our 14 days circadian experiment. (b) Larva locomotion (wildtype *D. melanogaster*) over 100 second captured with our system with 1 fps.

A similar distribution can be observed when our pipeline is employed on a system with a GPU, even though the average processing time of the temporal association is a bit higher between 134.54 and 154.11 ms. This amounts to an 48.0% share of runtime when used with YOLOv7-tiny compared to 28.7% when used with YOLOv7-E6E.

In a second step of our postprocessing routine we apply a centered median filter of size five to the associated developmental stage at each point in time for each individual specimen. This way, our tracker filters out single misclassifications that can appear on a frame-by-frame basis due to different factors, e.g. another object covers up the specimen or during the process of a larva pupating. Subsequently, after postprocessing, the determined positions and developmental stages over the time of the experiment for each specimen is saved in an output file in HDF5 that can directly be converted to be used for analysis in other established frameworks such as rhetomics [23] or custom analysis scripts.

**Performance evaluation.** To assess the performance of the Hatching-Box, we performed experiments with four different genotypes, a wild type line *iso31* [24] and the three well-known mutants for the period of emergence of adult flies [3]: *per^short^*, *per^long^*, *per^0^*. While wild type flies have a period of eclosion of about 24 hours in constant darkness, this behavior can be affected by mutations of the *period* gene. *per^short^* flies show an eclosion period of 19h, *per^long^* change it to 26h and *per^0^* fail to form a rhythmic pattern.

The experiments were performed in a temperature controlled incubator (CLF, Plant Climatics) set at 25°C, in constant darkness to be able to study the inner rhythm of the flies without effect of the light. We monitored three vials of each genotype per Hatching-Box with a image capturing interval of 10 minutes and configured the tracker to only track full and empty pupae. With this configuration we monitored the different genotypes for 14 days each and computed the quantitative results as described before in the former Subsection. To find the time points of eclosion/pupation based on this output we processed the sequence of detected developmental stages for each separate specimen with a sliding window of size $\tau$ (with $\tau = 7$ for our framerate). We mark a time point $t$ as time of eclosion/pupation if a majority of time points $[t, ..., t + \tau - 1]$ are associated with the class of empty/full pupa. For the performed circadian experiments we used this approach to extract the eclosion time points only.

In Fig 5 we can observe in the diagrams that at the beginning there is no eclosion, as expected, since most of the animals are in the stage of larvae. At about day 4-5 we start observing an increasing number of events. The *iso31* flies show a rhythm of 24 hours as expected, that is also detected in the periodogram, while in the *per^0^* there is no clear rhythm. The *per^long^* flies show a rhythm of 28.3 h, and interestingly, the beginning of the eclosion is delayed compared to the other genotypes, reflecting the importance of the circadian clock in regulating development [4]. Lastly, even though the periodogram of *per^short^* flies does not show any significant rhythm, it can be recognized in the in Fig 5a a short rhythm that is also detectable in the periodogram of about 19h, that would represent the published data.

## Materials and methods

### Hardware

**Hardware Design.** As the main compartment for our system we used an off-the-shelf electrical box measuring 355.0mm ×248.0mm ×110.7mm *(length ×width ×height)*, providing enough room for the necessary electrical components as well as three rearing vials with a diameter of 41.5mm (see S5 Fig). For our system we employed an adapted set of imaging components, backlight illumination and additional sensors as specified below.

**Imaging components.** Image recording is performed with the Raspberry Pi High Quality Camera module connected to the Raspberry Pi 4 (8 GB). The camera module uses the 12.3 megapixel Sony IMX477 sensor and is equipped with a C/CS mount which provides compatibility with a large number of camera lenses. When the camera is used with its full resolution of 4056 ×3040 pixels, our system can achieve a frame rate of up to 1 frame per second. For our

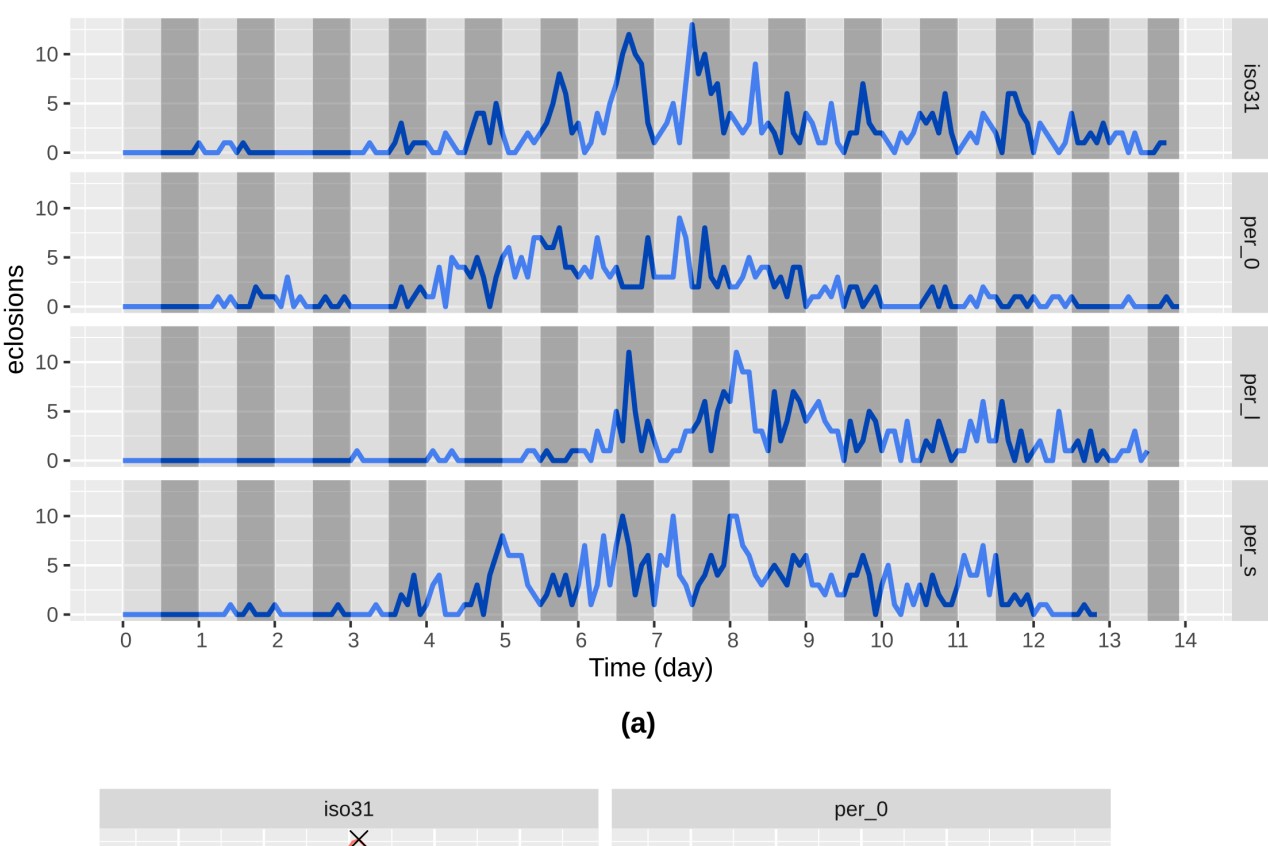

**(a)**

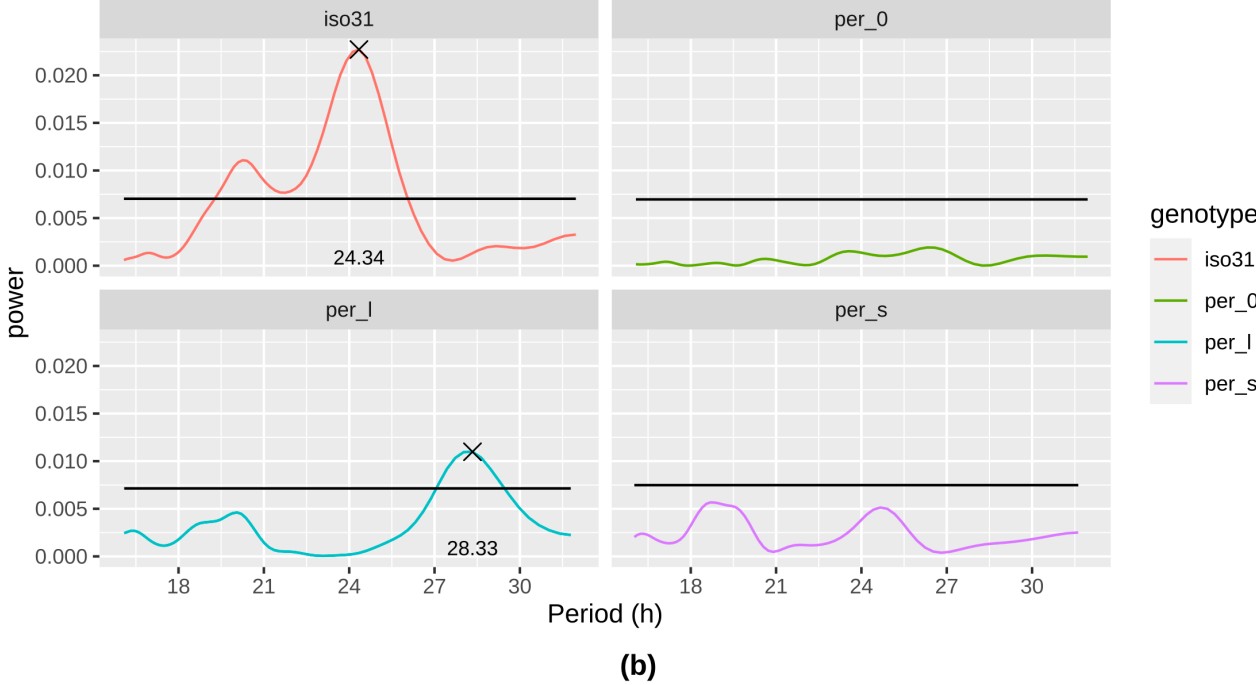

**(b)**

**Fig 5. Comparison of periodicity of selected clock mutants.** We used $per^0$, $per^{short}$, $per^{long}$ and wildtype *iso31* flies. For the statistical methods the R package *rhetomics* was used. (a) Eclosion events aggregated into 2 hour slices for the different genotypes over time. (b) Lomb-Scargle periodogram of observed mutants [25].

experiments we used the C125-0818-5M Basler lens with 8mm focal length due to its minimal working distance (MOD) of 100mm. We outfitted the lens with a MIDOPT LP818-46 low-pass IR filter with a useful range of 825nm - 1100nm. The builtin IR block filter of the camera module was removed.

**Backlight illumination.** Since the lighting conditions of an image have a significant influence on its characteristics and in turn on the performance of object detection algorithms we also propose a custom light guide panel design for the Hatching-Box. Our light guide panel is made of acrylic glass with punctures on the surface, distributing the incoming light homogeneously across its area.

The Hatching-Box uses two separate light sources. The first one consists of two LED strips emitting white light (250nm - 800nm) and can be used as a stimulus for the monitored *D. melanogaster*. A second LED strip consisting of IR LEDs with a wavelength of 890nm has been placed at the bottom edge of the light guide panel behind the rearing vial holder to provide a homogeneous illumination across its entire area. As a driver for the LEDs in our system, we use a standard Arduino Uno in combination with a custom-made shield that is connected to the Arduino's GPIO pins [26]. In their publication, Ogueta et al. used LED strips with four color channels red, green, blue and white (RGBW) which are connected to the shield. In our implementation we use three of these four channels for controlling the two white LED strips and the IR LED strip independently.

**Sensors.** Environmental conditions play a significant role for breeding and experimental designs. We therefore equipped the Hatching-Box with sensors for temperature and humidity (HS3003), light intensity (APDS9960) and barometric pressure (LPS22HB), all provided by the Arduino Sense BLE 33 sensor board. As an external digital light sensor to measure brightness outside of the Hatching-Box we use a BH1750. We implemented a custom-made firmware to query the Arduino Sense via serial connection which automatically provides all current measurements every time an image is taken and saved in the experiment's output HDF5 file.

## Software

**Overview.** Our software architecture consists of two main components, namely the headless Hatching-Box client application (HB-client) executed on the Raspberry Pi on each box independently and the Hatching-Box server application (HB-server), designed to be used on a central computer (see Fig 6). The HB-client controls the camera and lights of the Hatching-Box and provides access to the previously discussed integrated sensor board. On the other hand, the HB-server is executed on a computer connected to one or many Hatching-Box systems and provides an easy-to-use graphical user interface to control the capture modalities of each box. Moreover, the HB-server provides tools for long-term tracking and behavioral analysis of individual *D. melanogaster* specimen. For the purpose of communication, each Hatching-Box must be connected to the same network used by the server. Transmission of controls, images and other data is performed by a TCP/IP-based application protocol. By building our distributed architecture upon standard network protocols, our system can easily be extended or integrated into existing infrastructure.

**Client architecture: HB-client.** In our proposed architecture, the term "client" corresponds to an individual Hatching-Box . The soft- and hardware of each Hatching-Box is controlled by a Raspberry Pi 4. The Raspberry Pi is connected to a Raspberry Pi HQ camera module and, via USB, to an Arduino Sense BLE 33 sensor board which are used for capturing images and sensor data respectively. As a driver for the camera, the HB-client uses the libcamera software stack [27]. Libcamera enables control of the camera's recording parameters,

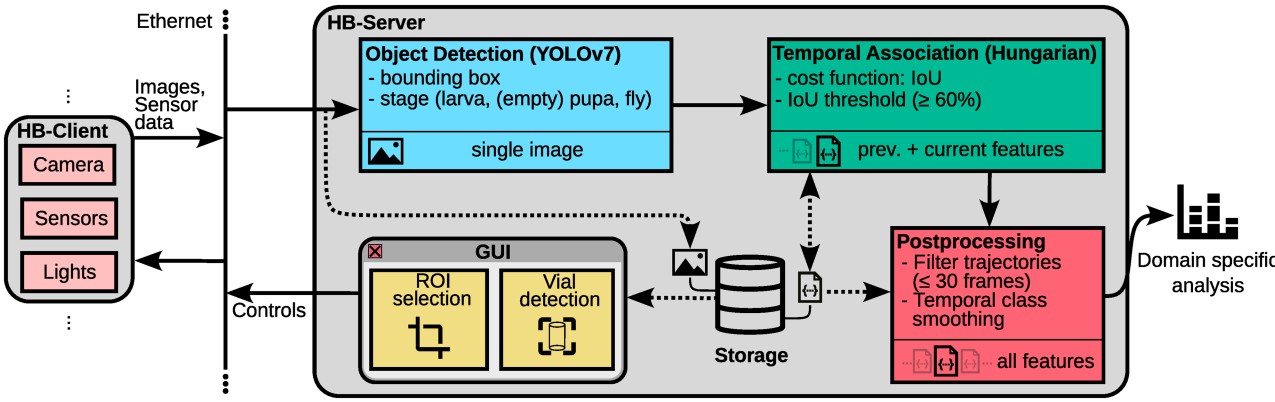

**Fig 6. Overview of data and command flow in our proposed system's architecture.**

e.g. exposure time and analog/digital gain, and construction of an image processing pipeline that uses firmware-based image processing in combination with custom-made processing routines. This processing includes routines for isolating a region of interest (ROI) from the captured image as well as detecting the boundaries of each monitored vial to support experimental designs which require observing different classes such as varying phenotypes. The parameters of the processing steps can be controlled in the HB-server application . Images are recorded using the BGR888 pixel format, encoded and compressed in PNG format and subsequently sent to the server. The LED controller, also connected via USB, is specifically designed for circadian experiments and operates the backlight panel. The connected LED strips with different wavelengths (here: white and IR LEDs) can be controlled separately. The HB-client is launched automatically upon startup of the Raspberry Pi and first initializes the serial communication channels to the sensor board, the LED controller and the libcamera software-stack. After initialization, the HB-client connects to the configured IP address and port of the HB-server. Afterwards, the Hatching-Box maintains the connection to the server, reconnects if necessary, and idles until the user issues a snapshot/recording task, changes camera settings or until the next scheduled video frame has to be captured and transmitted. By transferring the recorded images directly to the server, we are able to fully omit write operations to the Raspberry Pi's micro SD card, preventing data corruption.

**Server architecture: HB-server.** Following our architecture's naming scheme, the "server" is represented by a single, off-the-shelf computer, preferably with a GPU, which executes the HB-server application. The HB-server consists of three main components:

1. a TCP server,
2. a graphical user interface (GUI) and
3. an object tracker (see Fig 6).

For TCP communication the HB-server builds on the ZeroMQ library [28] to maintain reliable, scalable and concurrent network connections to each Hatching-Box simultaneously while minimizing the overhead.

The HB-server uses the Qt5 framework to provide a graphical user interface (GUI) to supervise and coordinate experiments. All online Hatching-Boxes in the same network are enumerated by its time of connection and accessible in the GUI. By selecting an individual Hatching-Box, users can control recording parameters and define a region of interest to select

specific vials to be monitored by the system. Previously recorded images and sensor readings (temperature, humidity, air pressure, light intensity and color) of the individual box can be reviewed. Multiple Hatching-Boxes and their adjusted settings can be organized in a *workspace*, which can be saved and than reused at a later time.

**Tracking.**  To analyse the images recorded by the HB-client we implemented an adapted tracking algorithm, which is accessible in the HB-server. This algorithm firstly separates the image into individual rearing vial crops using a one-time calibration step, which automatically generates a mask for each vial (see S2 Appendix). The subsequent tracker pipeline consists of two main steps:

1. detection and classification of objects in the crops over time and
2. association of the detected objects over time to compute consistent identities.

Detection and classification is done by the YOLOv7 framework, which has been used successfully on a wide range of domains including tiny insect detection [29] and medical imaging [30] and requires only moderate computing capabilities. For our experiments we used the baseline YOLOv7 model pretrained on the COCO dataset [31] and finetuned it on our custom Hatching-Box dataset . Loading and inference of our model is performed using the ONNX [32] framework to make use of platform-dependent acceleration methods. This decreases the inference time and provides an interface which makes it possible to also load other models to use in HB-server if desired. To further speed up the inference and the tracking process off-the-shelf GPU computing hardware is recommended.

The second step of our tracker is concerned with the temporal association and identity conservation in time. Assuming slow/non-moving objects (larvae or pupae) or an appropriate frame rate, the intersection-over-union (IoU) of bounding boxes of consecutive frames can be used as a metric for temporal association. The IoU is of two bounding boxes $b_i$ and $b_j$ is defined by

$$J(b_i, b_j) = \frac{|b_i \cap b_j|}{|b_i \cup b_j|}. \tag{1}$$

Based on the pairwise IoU values of all bounding boxes in frame $t$, $b_i^t$ $(i = 1, \ldots, m)$, and bounding boxes $b_j^{t+1}$ $(j = 1, \ldots, n)$ in the subsequent frame, we construct $I_{t,t+1} \in \mathbb{R}^{m \times n}$ such that

$$I_{t,t+1}(i,j) = J(b_i^t, b_j^{t+1}). \tag{2}$$

We use the hungarian algorithm [33] to find a bijective mapping between the bounding boxes in frame $t$ and frame $t + 1$ that maximizes

$$\sum_{i=1}^{m} \sum_{j=1}^{n} I_{t,t+1}(i,j). \tag{3}$$

To also allow detections that have no correspondence across both frames, additional $n$ rows and $m$ columns of dummy entries are added to $I_{t,t+1}$ to allow addition/deletion of objects over time. This approach is for example used for graph matching applications [34]. The resulting matrix $I_{t,t+1}$ has shape $n + m \times m + n$:

$$I_{t,t+1} = \begin{pmatrix} J(b_1^t, b_1^{t+1}) & J(b_1^t, b_2^{t+1}) & ... & J(b_1^t, b_n^t) & 1 & ... & 1 \\ J(b_2^t, b_1^{t+1}) & J(b_2^t, b_2^{t+1}) & ... & J(b_2^t, b_n^t) & 1 & ... & 1 \\ \vdots & \vdots & \ddots & \vdots & \vdots & \ddots & \vdots \\ J(b_m^t, b_1^{t+1}) & J(b_m^t, b_2^{t+1}) & ... & J(b_m^t, b_n^t) & 1 & ... & 1 \\ 1 & 1 & ... & 1 & 0 & ... & 0 \\ \vdots & \vdots & \ddots & \vdots & \vdots & \ddots & \vdots \\ 1 & 1 & ... & 1 & 0 & ... & 0 \end{pmatrix} \qquad (4)$$

Matching dummy objects with each other is prevented by setting $I_{t,t+1}(i,j) = 0$ for $m < i \leq m + n$ and $n < j \leq n + m$. Detected objects can be shown as an overlay on top of the captured images in the HB-server GUI.

### Circadian experiment

About 30 males and 30 females were crossed and their eggs collected in three vials (41.5 mm diameter) with standard yeast-containing fly food and kept at 25 C in a 12 h : 12 h light-dark cycles. The same parental flies were flipped twice, each time after 2 days of egglaying. The tubes without the parents were then directly put in the experimental setup for further studies. We excluded the recording of the early stages of larvae (L1 and L2) that stay mainly in the food–therefore only visible occasionally to our system–and studied only the wandering third instar larvae, which crawl out of the food looking for a place to pupate. The 4 genotypes used here (*iso31*, *per^short^*, *per^long^* and *per^0^*) have been described previously [3,24]. A set of three vials of the same genotype were tested simultaneously in the Hatching-Box. The experiment was performed at 25C in a temperature controlled incubator (CLF, Plant Climatics) in constant darkness.

### Discussion

Extracting large quantities of developmental and behavioral data is essential for a variety of biological experiments. However, the acquisition of this type of data involves tedious and time-consuming labor, aggravating the reproducibility and limiting the throughput. This predicament builds momentum for the emergence of automated hard- and software solutions to aid researchers gain new insights, e.g., in the field of neurosciences [35] and genetics [36]. However, the shift towards higher automatization also imposes new challenges and limitations which restrict the number of specimens that can be monitored and the length of the supported observation period. Most available systems focus on high temporal and spatial resolution when tracking behavioral features of animals such as their movement which renders them inappropriate for the analysis of long-term experiments or population development.

In this paper we have introduced an alternative tracking approach for long-term experiments. Our hardware and software combinations offers fully automatic developmental and behavioral quantifications, which are integrated into the regular rearing process, hence do not require any additional labor. The machine learning-based object detection and tracking algorithm enable the quantification of short larval trajectories, pupation and eclosion including their rhythmicity and temporal development as well as rudimentary activity monitoring. By making the exhaustive tracking data available as an HDF5 file, we additionally encourage

further analysis of the collected data with tools and frameworks already established by the user. The cost-effective and easy to build hardware, as well as the client-server-based software allow for arbitrary extensions yielding highly parallelized and high-throughput screenings.

The software is developed with a particular focus on throughput to detect, classify and track specimens in the captured images over the whole course of the experiment without manual intervention. The combination of these different components result in high quality tracking results for experiments with hundreds of animals spanning multiple weeks that are not straightforward to acquire with software-only tracking solutions (e.g. [15,19]) which require the experimenter to establish an appropriate imaging setup. The presence of user-controllable lights within each Hatching-Box facilitates multiple experiments with different light cycles in the same incubator. Additionally, by utilizing our sensor array, the environmental conditions during the experiment can be closely monitored and easily correlated with the obtained developmental data. Being able to track *D. melanogaster* in their rearing vials takes the burden off of experimenters to individualize their specimen and simultaneously makes the Hatching-Box also applicable for monitoring of their general development and gathering behavioral and fitness features on-the-fly.

Given our approach favors high throughput over fine-grain behavioral feature extraction, this introduces new challenges regarding identification of objects inside the rearing vials. On the one hand, off-the-shelf rearing vials are usually made of plastic, e.g., polypropylene, that is not as transparent as acrylic glass or certain other plastics depending on material quality which can complicate detection of animals. On the other hand, achieving a focal plane depth adequate for distinguishing animals in the cylindrical housing is not a trivial task. We mitigated this issue by choosing the focal depth in a way that objects in the camera-facing front of the rearing vial appear sharp and any other objects blurred so that foreground and background objects can be readily differentiated. Still, both the aforementioned factors can contribute to erroneous detections and classifications suggested by the YOLOv7 model. Nevertheless, our experimental results show that we are able to mitigate this issue by usage of our postprocessing routines which ultimately produce continuous trajectories even if detections are missing or inaccurate on a frame-by-frame basis.

In the future, we will integrate a vibration-free rotation mechanism for the rearing vials to further increase the total number of observable objects.

Due to framerate limitations of up to 1 frame per second (fps) when using the full 12 megapixel resolution of the camera module, our current implementation does not achieve a sufficient temporal resolution to track flying adult flies. We aim on further increasing the framerate to ultimately also allow behavioural analysis of fast moving specimen. This may pose further challenges, e.g., additional heat builtup and throughput limitations of the local network, which may involve modification of the current software architecture. Notably, at higher framerates, the storage of acquired images becomes impractical due to the substantial volume of generated data. To address this challenge, our system enables real-time processing of captured images, such that only the extracted behavioural metrics are retained for storage. However, given our focus on pupation and eclosion, we did not include this, as it falls outside the scope of the present study and will be investigated in future work.

## Supporting information

**S1 Fig. YOLOv7 evaluation.**
(PDF)

**S2 Fig. Double plotted actogram of *D. melanogaster* clock mutants and wildtype.**
(PDF)

**S3 Fig. Spatial distribution of wildtype *D. melanogaster* in larval and adult stage.**
(PNG)

**S4 Fig. Illustration of an exemplary pupation event as captured by our system.**
(PDF)

**S5 Fig. Technical drawings.**
(PDF)

**S6 Fig. Screenshot a the Hatching-Box server application (HB-server).**
(PNG)

**S1 Table. Bill of Material.**
(PDF)

**S1 File. Archive containing manufacturing files.**
(ZIP)

**S1 Appendix. Light guide panel design.**
(PDF)

**S2 Appendix. Automatic detection of vials.**
(PDF)

## Acknowledgments

We thank Ralf Stanewsky for providing the flies and for sharing laboratory space and equipment for our studies.

## Author contributions

**Conceptualization:** Julian Bigge, Benjamin Risse.

**Data curation:** Julian Bigge, Maite Ogueta, Luis Garcia.

**Formal analysis:** Julian Bigge, Maite Ogueta.

**Investigation:** Julian Bigge, Luis Garcia.

**Methodology:** Julian Bigge, Luis Garcia, Benjamin Risse.

**Project administration:** Benjamin Risse.

**Resources:** Benjamin Risse.

**Software:** Julian Bigge.

**Supervision:** Maite Ogueta, Benjamin Risse.

**Validation:** Julian Bigge, Maite Ogueta, Luis Garcia, Benjamin Risse.

**Visualization:** Julian Bigge.

**Writing – original draft:** Julian Bigge, Benjamin Risse.

**Writing – review & editing:** Julian Bigge, Maite Ogueta, Luis Garcia, Benjamin Risse.

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
