## [Decision Letter · Decision Letter 0]

24 Jun 2025

PONE-D-25-14944Hatching-Box: automated in situ monitoring of Drosophila melanogaster development in standard rearing vialsPLOS ONE

Dear Dr. Risse,

Thank you for submitting your manuscript to PLOS ONE. After careful consideration, we feel that it has merit but does not fully meet PLOS ONE’s publication criteria as it currently stands. Therefore, we invite you to submit a revised version of the manuscript that addresses the points raised during the review process.

DearThe current work deserved to be published provided to you improve the final version of your manuscript. Please ignore the "Bigge et al." the reviewer means "Risse et al"Good luck==============================

We look forward to receiving your revised manuscript.

Kind regards,

Rachid Bouharroud

Academic Editor

PLOS ONE

Journal Requirements:

2. We notice that your supplementary [figures/tables] are included in the manuscript file. Please remove them and upload them with the file type 'Supporting Information'. Please ensure that each Supporting Information file has a legend listed in the manuscript after the references list.

Reviewers' comments:

Reviewer's Responses to Questions

**Comments to the Author**

1. Is the manuscript technically sound, and do the data support the conclusions?

Reviewer #1: Yes

Reviewer #2: Partly

2. Has the statistical analysis been performed appropriately and rigorously? 

Reviewer #1: Yes

Reviewer #2: I Don't Know

3. Have the authors made all data underlying the findings in their manuscript fully available?

Reviewer #1: Yes

Reviewer #2: Yes

4. Is the manuscript presented in an intelligible fashion and written in standard English?

Reviewer #1: Yes

Reviewer #2: Yes

5. Review Comments to the Author

Reviewer #1: I have gone througout the manuscript entitled “Hatching-Box: automated in situ monitoring of Drosophila melanogaster development in standard rearing vials" (PONE-D-25-14944).” This manuscript is of great imprtance in lessening the timeconsumption and handling overcome by using machine learning via some rearing and romout and illumination algorithms. Moreover, the study of eclosion is still a common procedure to work the circadian rhythms of the flies, as well as the developmental timing of pupation. This type of experiments are time-consuming and often involve the constant personal monitoring of the rearing vials. The results propose the Hatching-Box, a novel in situ imaging and analysis system to automatically monitor and quantify the developmental behavior of Drosophila melanogaster in standard rearing vials and during regular rearing routines, rendering explicit experiments obsolete for automated monitoring in the general cultivation process.

This system enables custom tailored imaging hardware with dedicated detection and tracking algorithms, enabling the quantification of larvae, filled/empty pupae and flies over multiple days. The design of tracking algorithm and established circadian experiments by comparing the eclosion periods of wild type flies to the clock mutants pershort, perlong, per0 without involvement of any manual labor are well designed. The designed Hatching-Box is also able to extract additional information about group behavior as well as to reconstruct the whole life-cycle of the individual specimens. These results not only demonstrate the applicability of our system for long-term experiments. The importance of the automated monitoring and imaging system is well emphasized according to previously ones. However, some issues should be ascertained as follows:

1. Why were this automated system developed for circadian experiment by using the three D. melanogaster clock mutants pershort, perlong, per0 or why were these mutants used. As mentioned in the end of the abstract, can this system used for automated monitoring in the general cultivation process for example for rearing D. melanogaster or other insects for physiological or toxicological studies?

2. It was stated that each model system houses up to three rearing vials (41.5mm diameter), can be placed in incubators or cultivation rooms. In our lab, hundreds vials are kept in a incubators by means of shelfs. By this monitoring system and using illumination how can we track many of larvae and puape in one vials between others. Because, achieving a focal plane depth adequate for distinguishing animals in the cylindrical housing is not a trivial task as stated in the ms. Please do indicate this issues in the introduction section.

I am pleased to inform that this paper can be accepted fro puplication after a minor revision depends on above two minor issues

Reviewer #2: Bigge et al have developed a new Drosophila monitoring tool, which they call the Hatching-Box, that allows for automatic monitoring over multiple days in standard fly rearing vials. The device and software have been calibrated to detect different developmental stages (including embryo, larva, pupa and adult) with relatively high fidelity, identify transitions between these stages, and track individual larval movements. The main advantage of their device compared to currently existing systems is that image collection can be done in standard fly rearing vials and across multiple developmental stages. As such, it will be of interest to Drosophila researchers looking for automated tracking, particularly to determine the timing of eclosion events. There are several issues that should be addressed:

Major Points:

1. The authors say that the Hatching-Box offers the ability to “reconstruct the whole life-cycle of individual specimens”. However, they have not provided data confirming that this is the case. They have shown that they can detect with fairly high accuracy the different stages at single time points (Fig. 4), but they haven’t shown the ability to track a single individual across the life cycle. Do they have evidence of tracking flies over multiple developmental transitions in a single experiment, which would enable them to reconstruct the life cycle? Comparatively little space is used to discuss tracking of larval movements. The one figure (Fig. 3) shows only 100 s of data. It is also indicated that the frame rate has to be quite high to track larval movement, making long-term locomotor tracking unfeasible with this system because of the amount of data required to maintain necessary temporal resolution. Another issue is that the single plane of focus would not allow for tracking if larvae are moving to another focal plane. The ability to track larvae crawling in the food is also not discussed. For these reasons, is it not clear that a single individual could be followed across developmental stages in a vial containing many flies. The authors should provide proof of principle that this can be done or else tone down the claims made in the paper about the functions served by the Hatching-Box. Even without this capability, the Hatching-Box could be of benefit to researchers. It would be nice to see validation of at least one more specific use other than timing eclosion events. For example, can pupation events be tracked and quantified (which could be demonstrated through analysis of existing datasets)?

2. The authors have most convincingly shown the utility of Hatching-Box to track eclosion rhythms. Existing technologies, such as the Trikinetics Eclosion Monitor, also accomplish this task. The ability of the Hatching-Box to track eclosion in rearing vials is a definite upgrade on other methods. However, I would like to see how the Hatching-Box compares to established methods in other metrics. Based on the data from Figure 5, it seems like the detected rhythms aren’t as robust as those previously reported (eg by Konopka and Benzer). Is there a reason that this is the case? I would also propose that the authors perform manual eclosion tracking from at least one of the Hatching-Box analyzed videos to see how the Hatching-Box compares to a human observer.

3. There are no details provided on analysis of circadian eclosion rhythms (Fig. 5), apart from the mention of a Lomb-Scargle periodogram in the figure legend. Which days were used for analysis, how many etc? For the top ‘actograms’ shown in Fig. 5, is this data from one representative vial? This should be included in the methods section of the manuscript. Furthermore, in lines 248-251 they say “Lastly, even though the periodogram of pershort flies does not show any significant rhythm, it can be recognized in the actogram a short rhythm that is also detectable in the periodogram of about 19h, that would represent the published data.” I do not see a short period phenotype in the data in Fig. 5. It would help if the authors could indicate on the graph where they think the data show the short period rhythm, or graph and analyze the data in a different way that supports their conclusion.

4. Fig. 2 is referenced after Fig. 1 in the text. There is no text reference to Fig. 3. It would be helpful to include written references to all figures and to reference them in the order they appear.

5. Often acronyms are used but not spelled out in the text. Eg, in lines 144-146: “The best performing model shows 4.35% better mAP on 50% to 95% IoU compared to our YOLOv7-tiny baseline, while it only shows a surplus of 0.73% in mAP on 50% IoU (see Fig. 4).” For a non-expert, it is necessary to state what mAP and IoU indicate.

Minor Points:

1. The abstract claims that the Hatching-Box will “automatically monitor and quantify the developmental behavior of Drosophila melanogaster in standard rearing vials and during regular rearing routines, rendering explicit experiments obsolete”. This latter claim is quite hyperbolic. Crosses must still be set and timed, with appropriate control lines. Furthermore, there are several inherent limitations to using the hatching box that might make additional experiments necessary. I would tone down the assertion that the device will make explicit experiments obsolete.

2. The introduction mentions that since 2023, “flyGear offers a commercially available system that can be used to classify D. melanogaster larvae and pupae”. Is there a citation available for this company or technology? I could not find anything with a google search.

3. It would be useful to show some screenshots of the GUI so that the reader could get an understanding of the user interface and the functions it provides.

6. PLOS authors have the option to publish the peer review history of their article (what does this mean?). If published, this will include your full peer review and any attached files.

Reviewer #1: **Yes: **Prof. Dr. Kemal Büyükgüzel

Reviewer #2: No

---

## [Author Response · Author response to Decision Letter 1]

15 Aug 2025

Dear Rachid Bouharroud,

We thank the reviewers for their thoughtful and constructive feedback, which has allowed us to further refine the focus and clarity of our study. Please find our detailed responses to the points raised below.

**Applicability of our System for other Organisms and Studies**

As a proof-of-concept, we decided to chose the circadian experiment with the different per mutations of D. melanogaster as there is comprehensive literature about this type of study. Even though our system was mainly designed with fruit flies in mind, generally the system is also able to monitor other species. However, depending on the dissimilarity compared to Drosophila of the species to study, the YOLO object detector might have to be retrained on images of the new species to accurately detect and classify the specimens. Furthermore, while we have not yet explicitly tested the system in a toxicological context, its performance in detecting overall behavioural patterns and developmental trends suggests that it should be well-suited for such experiments. The ability to monitor and quantify changes in population-level activity and development enables sensitive detection of toxicological effects. We plan to explore and validate these applications in future work.

**Claim Regarding Whole Lifecycle Reconstruction and Focus on Eclosion Events**

We appreciate the reviewer’s observation that our current system does not demonstrate full reconstruction of the entire life cycle, and instead focuses on eclosion events. Even though live-long monitoring of the population in the vial is supported and objects are continuously classified into larvae, pupae and adults, we do not track complete life cycles of individual animals. To emphasis this fact and to avoid misunderstanding we revised the manuscript and adapted all claims and terminology accordingly.

Revised Focus: Our system aims to deliver an optimal trade-off between hardware complexity, usability, and feature extraction. The main goal is to enable convenient and robust population-level activity monitoring rather than comprehensive life history tracking for each individual. We have emphasized population development and gross behavioral and developmental trends, rather than individual trajectories in our manuscript.

To underscore this, we have added additional data visualizations in the form of distribution heatmaps for both larval and adult (fly) stages using our YOLO-based detection to the Supplementary Material. These maps demonstrate the system's utility for monitoring overall population activity and spatial distributions, which aligns with our intended application.

**Larvae Tracking: Technical and Conceptual Limitations**

High Framerate & Data Management: We acknowledge that real-time high-framerate tracking of larvae is currently infeasible within our system, primarily due to data storage and computational limits. While real-time (on-the-fly) processing could be a potential workaround, development and testing of such solutions are beyond the scope of the current manuscript. We have clarified in the revised text that current results are intended as a proof-of-concept to illustrate the system's potential, rather than as evidence for full lifecycle tracking.

Section on Larvae Tracking: In line with the above, we have also adjusted the relevant sections concerned with the tracking of larvae to more explicitly state these limitations and the proof-of-concept nature of our demonstration.

Loss of Larvae Out of Focal Plane: We agree this is an inherent limitation of single-camera systems – larvae may move out of the focal plane and be temporarily or permanently lost. Although technical improvements such as rotating vials or multiple cameras could mitigate this, our design prioritizes simplicity and throughput to enable high levels of parallelization. For population-level monitoring, loss of individuals may be compensated statistically by the increased number of tracked larvae. Long-term identity-preserving tracking is outside our scope. To mitigate this challenge in our current implementation, our detection model is trained to recognize and classify out-of-focus larvae as a separate class. We therefore keep track of in and out of focus objects (helps maintain total population counts) while excluding explicit classifications into pupae or larvae, which is often impossible given the inferior contrast of non-focused objects.

We added this information to the manuscript.

Larvae Crawling into Food Not Covered: Thank you for pointing out this gap. We have expanded the Methods section to acknowledge that larvae may spend significant time within food, and as such are undetectable in our setting and clarified our focus on ``wandering stage'' larvae, as monitoring those outside the food is both practical and consistent with certain biological interests in this developmental stage.

**Inclusion of Pupation Events**

We agree that including pupation transitions would further strengthen the population-development perspective.

We included an explanation why detection of pupation is less robust using visual data only, as the behavioral and morphological changes are more gradual and rely on indirect proxies (e.g., immobility) in our revised manuscript. We have also included a novel figure showing a representative pupation event of a wiltype larva as captured by our system (Suppl. Fig. S4). The transition from larvae to pupae can clearly be seen, however given the visually continuous nature of this event, the exact moment of pupation is less clear than eclosion.

**Robustness and Variability Compared to Konopka & Benzer**

The reviewer notes that our eclosion period results (Fig. 5) appear less robust and ``clean'' than those of Konopka & Benzer. After reviewing their methods, we note that their data was acquired at a lower temporal resolution of one hour and hand-counted, inherently leading to smoother, less variable curves. Our automated, camera-based system is capable of higher temporal granularity, which, while capturing more detail, can also surface biological or technical variability previously unreported. A similar effect can be observed when comparing the results of the WEclMon system with the TriKinetics systems (compare to Fig. 2 to in [11]).

In accordance with the approach described by Konopka & Benzer, we have revised our periodicity visualizations by aggregating the data from the 10-minute intervals provided by our system to 1-hour intervals. This adjustment enhances the clarity and statistical power of the observed periodic patterns, and results in visualizations that are more closely aligned with the originally published data, while still preserving the period lengths reported by our system. The updated visualization is attached below and can also be integrated in the supplement of our manuscript.

Furthermore, we appreciate the reviewer’s suggestion to compare our automatic tracking method with manual labeling. We would like to clarify that our automatic detection algorithm was trained and validated on manually labeled eclosion events, effectively ensuring that its performance was benchmarked against manual annotations. As a result, the accuracy of our automatic method has already been directly assessed through this comparison as part of our model development process. We have now stated this more explicitly in the Dataset section of our revised manuscript.

**Actogram/Periodogram (Fig. 5) Support for Rhythmicity Conclusions**

Thank you for suggesting further analyses and clearer annotation. We have added clear information on the dates, duration, and number of vials analyzed. To facilitate interpretation, we have explored alternative visualization methods to more clearly illustrate periodicity. In particular, we have added a line graph depicting the number of eclosions across consecutive days of our experiments. Additional text has been included in the Results and Methods sections to clarify the experimental duration and conditions (flies monitored for 2 weeks post-oviposition, with eclosion events occurring from days 2--4, and after parental removal).

**Summary**

We hope these clarifications and additional analyses address the reviewers' concerns and improve the focus and rigor of our manuscript. Above all, we have revised our claims to emphasize population-level activity monitoring and the balance between system complexity and usability, rather than comprehensive individual tracking. Additionally, as requested by the reviewer, we also added a screenshot of our software to the Supplements.

Thank you once more for your valuable input.

---

## [Editor Report · Decision Letter 1]

19 Aug 2025

Hatching-Box: automated in situ monitoring of Drosophila melanogaster development in standard rearing vials

PONE-D-25-14944R1

Dear Dr. Risse,

We’re pleased to inform you that your manuscript has been judged scientifically suitable for publication and will be formally accepted for publication once it meets all outstanding technical requirements.

Kind regards,

Rachid Bouharroud

Academic Editor

PLOS ONE
---

## [Editor Report · Acceptance letter]

PONE-D-25-14944R1

PLOS ONE

Dear Dr. Risse,

I'm pleased to inform you that your manuscript has been deemed suitable for publication in PLOS ONE. Congratulations! Your manuscript is now being handed over to our production team.

Kind regards,

on behalf of

Dr. Rachid Bouharroud

Academic Editor

PLOS ONE